# DEEP PROBABILISTIC SUBSAMPLING FOR TASK-ADAPTIVE COMPRESSED SENSING

**Iris A.M. Huijben**[*]
Department of Electrical Engineering
Eindhoven University of Technology
Eindhoven, The Netherlands
`i.a.m.huijben@tue.nl`

**Bastiaan S. Veeling**[*]
Department of Computer Science,
University of Amsterdam
Amsterdam, The Netherlands
`basveeling@gmail.com`

**Ruud J.G. van Sloun**
Department of Electrical Engineering
Eindhoven University of Technology
Eindhoven, The Netherlands
`r.j.g.v.sloun@tue.nl`

## ABSTRACT

The field of deep learning is commonly concerned with optimizing predictive models using large pre-acquired datasets of densely sampled datapoints or signals. In this work, we demonstrate that the deep learning paradigm can be extended to incorporate a subsampling scheme that is jointly optimized under a desired sampling rate. We present Deep Probabilistic Subsampling (DPS), a widely applicable framework for task-adaptive compressed sensing that enables end-to-end optimization of an optimal subset of signal samples with a subsequent model that performs a required task. We demonstrate strong performance on reconstruction and classification tasks of a toy dataset, MNIST, and CIFAR10 under stringent subsampling rates in both the pixel and the spatial frequency domain. Thanks to the data-driven nature of the framework, DPS is directly applicable to all real-world domains that benefit from sample rate reduction. The code used for this paper is made publicly available[1].

## 1 INTRODUCTION

In many real-world prediction problems, acquiring data is expensive and often bandwidth-constrained. Such is the case in regimes as medical imaging (Lustig et al., 2007; Choi et al., 2010; Chernyakova & Eldar, 2014), radar (Baraniuk, 2007), and seismic surveying (Herrmann et al., 2012). By carefully reducing the number of samples acquired over time, in pixel-coordinate space or in k-space, efficient subsampling schemes lead to meaningful reductions in acquisition time, radiation exposure, battery drain, and data transfer.

Subsampling is traditionally approached by exploiting expert knowledge on the signal of interest. Famously, the Nyquist theorem states that when the maximum frequency of a continuous signal is known, perfect reconstruction is possible when sampled at twice this frequency. More recently, it has been shown that if the signal is sparse in a certain domain, sub-Nyquist rate sampling can be achieved through compressive measurements and subsequent optimization of a linear system under said sparsity prior; a framework known as compressed sensing (CS) (Donoho et al., 2006; Eldar & Kutyniok, 2012; Baraniuk, 2007).

CS methods however lack in the sense that they do not (under a given data distribution) focus solely on the information required to solve the downstream task of interest, such as disease prediction or semantic segmentation. Formalizing such knowledge is challenging in its own right and would require careful analysis for each modality and downstream task. In this work, we propose to explore

---

[*]Equal contribution.
[1]`https://github.com/IamHuijben/Deep-Probabilistic-Subsampling.git`

the deep learning hypothesis as a promising alternative: reducing the need for expert knowledge in lieu of large datasets and end-to-end optimization of neural networks.

As subsampling is non-differentiable, its integration into an end-to-end optimized deep learning model is non-trivial. Here we take a probabilistic approach: rather than learning a subsampling scheme directly, we pose a probability distribution that expresses belief over effective subsampling patterns and optimize the distribution's parameters instead. To enable differentiable sampling from this distribution, we leverage recent advancements in a continuous relaxation of this sampling process, known as Gumbel-softmax sampling or sampling from a concrete distribution (Jang et al., 2017; Maddison et al., 2016). This enables end-to-end training of both the subsampling scheme and the downstream model.

Naively, the number of parameters of a distribution over an n-choose-k problem scales factorially, which is intractable for all practical purposes. We propose a novel, expressive yet tractable, parameterization for the subsampling distribution that conditions on the output sample index. We hypothesize that such conditioning prevents redundant sampling: it enables modeling the scenario in which multiple candidate samples can be equally good, yet redundant in combination. Furthermore, we investigate a parameter-restricted approach with a single parameter per candidate sample, balancing tractability and exploration (Kool et al., 2019). In this case, we adopt a continuous relaxation of top-K sampling to guarantee differentiability (Plötz & Roth, 2018).

Our main contributions are as follows:

- DPS: A new regime for task-adaptive subsampling using a novel probabilistic deep learning framework for incorporating a sub-Nyquist sampling scheme into an end-to-end network.

- DPS enables joint optimization of a subsampling pattern with a predictive downstream model, without the need for explicit knowledge on a sparsifying basis.

- We demonstrate improved performance over strong subsampling baselines in image classification and reconstruction, while sampling both in Fourier and pixel space.

## 2 RELATED WORK

Recent works have proposed deep-learning-based subsampling methods for fast MRI, historically being one of the most prominent applications of CS. Weiss et al. (2019) exploit gradient backpropagation to a fixed set of real-valued coordinates, enabled by their subsequent (limited-support) interpolation on the discrete k-space grid, and Bahadir et al. (2019) formulate the sampling problem by learning pixel-based thresholding of i.i.d. samples drawn from a uniform distribution, referred to as LOUPE. Where the former suffers from limited exploratory capabilities (likely due to its compact support), with learned sampling schemes typically not deviating far from their initialization, the latter controls the sample rate only indirectly, through the addition of a sparsity-promoting $\ell_1$ penalty on the sampling mask.

Related to our methodology, yet different in purpose, Plötz & Roth (2018); Xie & Ermon (2019); Kool et al. (2019) leverage an extension to the Gumbel-max trick (Gumbel, 1954) for (ordered) subset selection using a categorical distribution with $N-1$ free parameters. They rely on top-K sampling, as first proposed by Vieira (2014). Based upon the continuous relaxation of the categorical distribution, known in the deep learning community as the Gumbel-softmax trick or the concrete distribution (Jang et al., 2017; Maddison et al., 2016), Plötz & Roth (2018) show that such a relaxation also exists for top-K sampling. In this work, we investigate the use of Gumbel-based relaxations for task-adaptive compressive sampling, and further propose a novel parametrization where we condition the sampling distribution on the output sample index.

We differentiate our contribution from deep encoder-decoder methods for data compression (Baldi & Hornik, 1989; Hinton & Zemel, 1993; Blier & Ollivier, 2018; Habibian et al., 2019), which do not aim at reducing data rates already at the sensing and digitization stage. Related work by Mousavi et al. (2019) and Wu et al. (2019), focusses on the problem of learning compressive linear encoders/filters, rather than discrete subsampling as addressed here. The authors of Balın et al. (2019) use a Gumbel-max-based auto-encoder for discrete feature selection, which is however not task-adaptive.

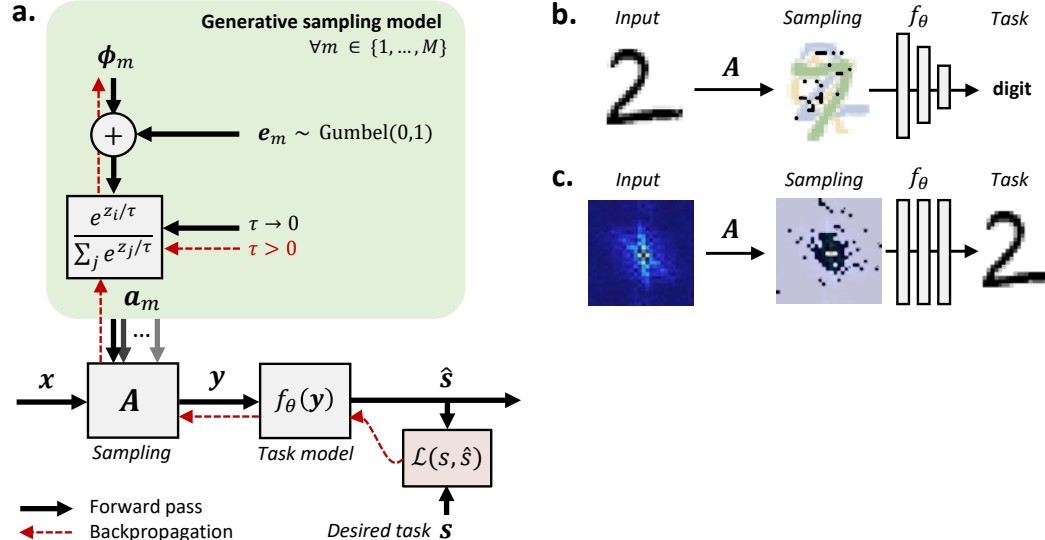

Figure 1: (a) System-level overview of the proposed framework, in which a probabilistic generative sampling model (DPS) and a subsequent task model are jointly trained to fulfill a desired system task. (b,c) Two illustrative task-based sampling paradigms: image classification from a partial set of pixels (b), and image reconstruction from partial Fourier measurements (c), respectively.

Through the lens of contemporary deep learning, subsampling can be interpreted as a form of *attention* (Bahdanau et al., 2014; Kim et al., 2017; Parikh et al., 2016; Vaswani et al., 2017). Rather than attending on intermediate representations, our model "attends" directly on the input signal. For subsampling to be effective, sparse weights are essential. In the space of attention, this is known as *hard* attention (Xu et al., 2015), and is typically optimized using the REINFORCE gradient estimator (Williams, 1992). In contrast to the method of attention as applied in these works, our method aims for a fixed, reduced subsampling rate.

## 3 METHOD

### 3.1 TASK-ADAPTIVE SYSTEM MODEL

We consider the problem of predicting some downstream task $s$, through a learned subsampling scheme $\mathbf{A}$ on a fully-sampled signal $\mathbf{x} \in \mathbb{R}^N$, resulting in measurements $\mathbf{y} \in \mathbb{R}^M$, with $M << N$:

$$\mathbf{y} = \mathbf{A}\mathbf{x}, \tag{1}$$

where $\mathbf{A} \in \{0,1\}^{M \times N}$ is the subsampling measurement matrix. We here concern ourselves specifically with scenarios in which the rows of $\mathbf{A}$ are constrained to having cardinality one, i.e. $||a_m||_0 = 1, \forall m \in \{1, ..., M\}$. In such cases, $\mathbf{A}$ serves as a subset selector, sampling $M$ out of $N$ elements in $\mathbf{x}$. From the resulting low-rate measurements $\mathbf{y}$, we aim at predicting task $s$ through:

$$\hat{s} = f_\theta(\mathbf{y}), \tag{2}$$

with $f_\theta(\cdot)$ being a function that is differentiable with respect to its input and parameters $\theta$, e.g. a neural network.

Given a downstream task and dataset, we are interested in learning both the optimal processing parameters $\theta$, and the sampling scheme as described in eq. (1). To circumvent the non-differential nature of discrete sampling, we will introduce a novel fully probabilistic sampling strategy that allows for gradient-based learning through error backpropagation, on which we detail in the following section.

### 3.2 DPS: DEEP PROBABILISTIC SUBSAMPLING

Since direct optimization of the elements in $\mathbf{A}$ is intractable due to its combinatorial nature, we here instead propose to leverage a tractable generative sampling model that is governed by a learned subsampling distribution, parameterized by $\boldsymbol{\Phi}$:

$$\mathbf{A}_{(\boldsymbol{\Phi})} \sim P(\mathbf{A}|\boldsymbol{\Phi}). \tag{3}$$

Thus, rather than optimizing $\mathbf{A}_{(\boldsymbol{\Phi})}$, we optimize the distribution parameters $\boldsymbol{\Phi}$. To warrant sufficient expressiveness while maintaining tractability, we learn the parameters $\boldsymbol{\phi}_m \in \mathbb{R}^N$ of $M$ independent categorical distributions (rather than the joint distribution, which scales factorially), being the rows of $\boldsymbol{\Phi} \in \mathbb{R}^{M \times N}$.

Formally, we define each $m^{\text{th}}$ measurement row $\mathbf{a}_m \in \{0,1\}^N$ in $\mathbf{A}_{(\boldsymbol{\Phi})}$ as a one-hot encoding of an independent categorical random variable $r_m \sim \text{Cat}(N, \boldsymbol{\pi}_m)$. We define $\boldsymbol{\pi}_m \in \mathbb{R}^N = \{\pi_{m,1}, \ldots, \pi_{m,N}\}$, being a vector containing $N$ class probabilities, and parameterize it in terms of its unnormalized logits $\boldsymbol{\phi}_m$ such that:

$$\pi_{m,n} = \frac{\exp \phi_{m,n}}{\sum_{i=1}^{N} \exp \phi_{m,i}}. \tag{4}$$

We sample from $\text{Cat}(N, \boldsymbol{\pi}_m)$ by leveraging the Gumbel-max trick (Gumbel, 1954), a reparameterization of the sampling procedure. It perturbs the unnormalized logits $\phi_{m,n}$ with i.i.d. Gumbel noise[2] samples $e_{m,n} \sim \text{Gumbel}(0, 1)$. The argmax of these perturbed logits, results in a sample from the original categorical distribution $\text{Cat}(N, \boldsymbol{\pi}_m)$. A realization $\tilde{r}_m$ is then defined as:

$$\tilde{r}_m = \underset{n}{\operatorname{argmax}}\{w_{m-1,n} + \phi_{m,n} + e_{m,n}\}, \qquad m \in \{1, \ldots, M\}, \tag{5}$$

in which $w_{m-1,n} \in \{-\infty, 0\}$ for $n \in \{1 \ldots N\}$ mask previous realizations $\tilde{r}_1, \cdots, \tilde{r}_{m-1}$, by adding $-\infty$ to the logit of the previously selected category, enforcing sampling without replacement among the $M$ distributions. Introducing the function $\text{one\_hot}_N(\cdot)$ as the operator that returns a one-hot vector of length $N$, we finally obtain:

$$\mathbf{a}_m = \text{one\_hot}_N\{\tilde{r}_m\} = \text{one\_hot}_N\left\{\underset{n}{\operatorname{argmax}}\{w_{m-1,n} + \phi_{m,n} + e_{m,n}\}\right\}. \tag{6}$$

To permit error backpropagation for efficient optimization of $\boldsymbol{\Phi}$, we require $\nabla_{\boldsymbol{\phi}_m}\mathbf{a}_m$ to exist $\forall m \in \{1, \ldots, M\}$. Since $\operatorname{argmax}(\cdot)$ is a non-differentiable operator, we adopt the Straight-Through Gumbel Estimator (Jang et al., 2017; Maddison et al., 2016) as a surrogate for $\nabla_{\boldsymbol{\phi}_m}\mathbf{a}_m$:

$$\nabla_{\boldsymbol{\phi}_m}\mathbf{a}_m := \nabla_{\boldsymbol{\phi}_m}\mathbb{E}_{\mathbf{e}_m}\left[\text{softmax}_\tau(\mathbf{w}_{m-1} + \boldsymbol{\phi}_m + \mathbf{e}_m)\right] =$$
$$\nabla_{\boldsymbol{\phi}_m}\mathbb{E}_{\mathbf{e}_m}\left[\frac{\exp\{(\mathbf{w}_{m-1} + \boldsymbol{\phi}_m + \mathbf{e}_m)/\tau\}}{\sum_{i=1}^{N}\exp\{(w_{m-1,i} + \phi_{m,i} + e_{m,i})/\tau\}}\right], \tag{7}$$

with (row operator) $\text{softmax}_\tau(\cdot)$ as a continuous differentiable approximation of the one-hot encoded $\operatorname{argmax}(\cdot)$ operation. Appendix A provides the full derivation of $\nabla_{\boldsymbol{\phi}_m}\mathbf{a}_m$, and fig. 1 shows a schematic overview of the proposed framework.

We refer to sampling using the $\text{softmax}_\tau(\cdot)$ function as soft sampling. Its temperature parameter $\tau$ serves as a gradient distributor over multiple entries (i.e. logits) in $\boldsymbol{\phi}_m$. Using a relatively high value enables updating of multiple logits during training, even though a hard sample was taken in the forward pass. In the limit of $\tau \to 0$, soft sampling approaches the one-hot encoded $\operatorname{argmax}(\cdot)$ operator in eq. (6) (Jang et al., 2017; Maddison et al., 2016). While lowering $\tau$ reduces the gradient estimator's bias, it comes at the cost of a higher variance. We find however that using a fixed (tuned) value of $\tau$ worked well for our experiments, and do not explore methods to reduce the introduced gradient bias (Grathwohl et al., 2017; Tucker et al., 2017).

---

[2]The Gumbel distribution is typically used to model the maximum of a set of independent samples. Its probability density function is of the form: $f(x) = \frac{1}{\beta}e^{-z-e^{-z}}$, with $z = \frac{x-\mu}{\beta}$, where $\mu$ and $\beta$ are the mean and standard deviation, respectively. The standard Gumbel distribution, with $\mu = 0$ and $\beta = 1$ is adopted in the Gumbel-max trick.

To study the trade-off between parameter efficiency and convergence, we further study a more parameter-restricted version of DPS that shares the weights $\phi_m$ across all $M$ distributions. This allows for a more memory-efficient implementation, by leveraging the Gumbel top-K trick (Kool et al., 2019) (rather than Gumbel-max), and its corresponding relaxation (Plötz & Roth, 2018). We refer to this particular model as DPS-topK, and adopt DPS-top1 to indicate the sampling model presented in equations 6 and 7, where weights are not shared across distributions, and one sample is drawn from each distribution. Algorithm 1 in appendix B describes the DPS algorithm.

## 4 EXPERIMENTS

We test the applicability of the proposed task-adaptive DPS framework for three datasets and two distinct tasks: image classification and image reconstruction. We explore subsampling in pixel-coordinate space as well as in k-space. The latter is relevant for scenarios in which data is acquired in the frequency domain, such as (but not limited to) magnetic resonance imaging (MRI).

### 4.1 MNIST CLASSIFICATION

**Experiment setup** Classification performance was tested on the MNIST database (LeCun et al., 1998), comprising 70,000 $28 \times 28$ grayscale images of handwritten digits 0 to 9. We split the dataset into 50,000 training images, 5,000 validation, and 5,000 test images. We train DPS-top1 and DPS-topK to take partial measurements in either the image or Fourier domain, and process them through the task model to yield a classification outcome. Results are compared to those obtained using uniformly distributed pixel/Fourier samples, a sampled disk/low pass filter, and the data-driven LOUPE method (Bahadir et al., 2019).

Besides, to show the benefit of task-adaptive sampling, i.e. jointly training sampling with the task, we add a case in which we disjointly learn the classification task from the sampling pattern. As such, we train DPS-topK for MNIST reconstruction, after which the learned sampling pattern is frozen, and the classifier is trained.

**Task model** After sampling $M$ elements, all $N$ zero-masked samples (or $2N$ in the case of complex Fourier samples) are passed through a series of 5 fully-connected layers, having $N$, 256, 128, 128 and 10 output nodes, respectively. The activations for all but the last layer were leaky Re-LUs, and 20% dropout was applied after the first three layers. The 10 outputs were normalized by a softmax function to yield the respective classification probabilities. Table 1 in appendix C shows a detailed overview of the complete architecture of the task model. Zero-filling and connecting all possible samples, rather than only connecting the $M$ selected samples, facilitated faster co-adaptation of the network to different sampling patterns during training.

**Training details** We train the network to maximize the log-likelihood of the observations $\mathcal{D} = \{(\mathbf{x}_i, \mathbf{s}_i) \mid i \in 0, \ldots, L\}$ through minimization of the categorical cross-entropy between the predictions and the labels, denoted by $\mathcal{L}_s$. We moreover promote training towards one-hot distributions $\mathrm{Cat}(N, \boldsymbol{\pi}_m)$ by penalizing high entropy, adopting:

$$\mathcal{L}_e = -\sum_{m=1}^{M}\sum_{n=1}^{N} \pi_{m,n} \log \pi_{m,n}, \tag{8}$$

with $\pi_{m,n}$ defined as in eq. (4). The total optimization problem is thus:

$$\hat{\boldsymbol{\Phi}}, \hat{\theta} = \underset{\boldsymbol{\Phi},\theta}{\mathrm{argmin}}\Big\{\mathbb{E}_{(\mathbf{x},\mathbf{s})\sim p_{\mathcal{D}}}\mathcal{L}_s + \mu\mathcal{L}_e\Big\}, \tag{9}$$

where $p_{\mathcal{D}}$ is the data generating distribution. Penalty multiplier $\mu$ was set to linearly increase $1e-5$ per epoch, starting from 0.0. Increasing this entropy penalty enforces one-hot distributions, and therefore less variations in the realizations, i.e. sampling patterns, when training evolves. The temperature parameter $\tau$ in eq. (7) was set to 2.0, and the sampling distribution parameters $\boldsymbol{\Phi}$ were initialized randomly, following a zero-mean Gaussian distribution with standard deviation 0.25. Equation 9 was optimized using stochastic gradient descent on batches of 32 examples, approximating the expectation by a mean across the train dataset. To that end, we used the ADAM solver

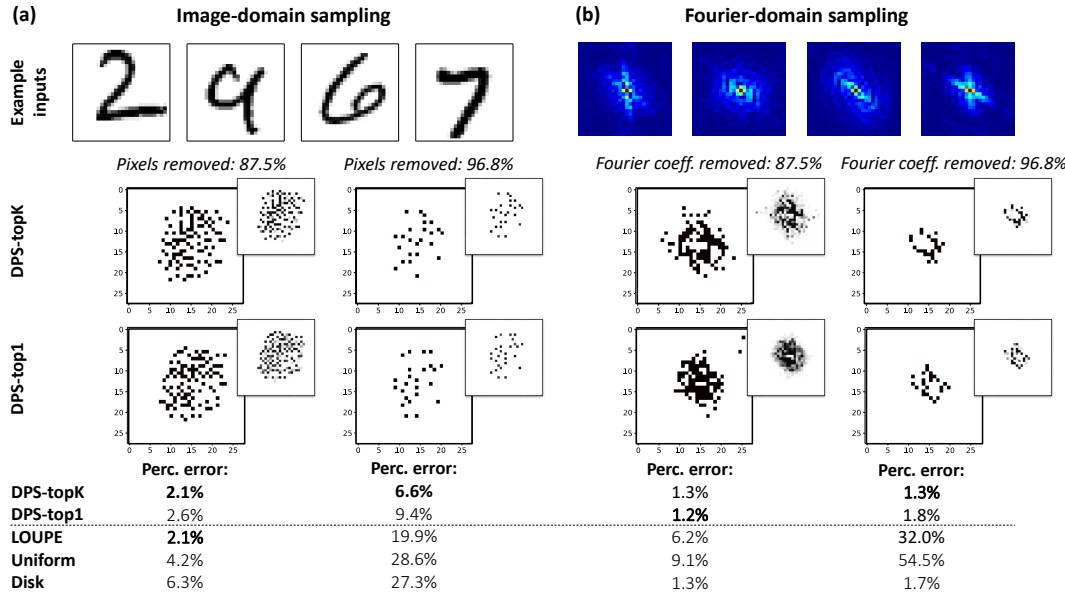

Figure 2: MNIST classification for (a) image-domain, and (b) Fourier-domain subsampling. (top) Several example images in the respective sampling domains, ($2^\text{nd}$ and $3^\text{rd}$ row) learned task-adaptive (DPS-topK and DPS-top1, respectively) subsampling patterns, with their relative sample incidence across a 1000 such realizations (inset), and (bottom) hold-out classification results of the proposed DPS methods, compared to the LOUPE baseline, and two non-learned baseline sampling approaches.

($\beta_1 = 0.9$, $\beta_2 = 0.999$, and $\epsilon = 1e-7$ ) (Kingma & Ba, 2014), and we trained until the validation loss plateaued. We adopted different learning rates for the sampling parameters $\Phi$ and the parameters of the task model $\theta$, being $2e-3$ and $2e-4$, respectively.

**Results** The results presented in fig. 2a show that image-domain sampling using DPS significantly outperforms the fixed sampling baselines (uniform and disk). The data-driven LOUPE method is outperformed in case of a strong subsampling factor. The resulting patterns qualitatively demonstrate how, for this task, a sensible selection of pixels that are most informative was made (slightly slanted, and capturing discriminative areas). Notably, partial Fourier measurements (fig. 2b) allowed for a much greater reduction of the number samples, with DPS sampling outperforming uniform sampling and LOUPE, and showing similar performance as fixed disk sampling. Interestingly, the DC and very-low frequency components were consistently not selected.

Figure 3 shows the learned sampling pattern in pixel space (96.8% pixels removed), when training DPS-topK for the reconstruction task. The adopted reconstruction network architecture is the same as the one used for CIFAR10 reconstruction (see section 4.3). The subsequently trained classifier resulted in a classification error of 9.3%, compared to 6.6% for task-adaptive learning with DPS-topK. This case clearly illustrates the benefit of task-adaptive learning of a sampling pattern.

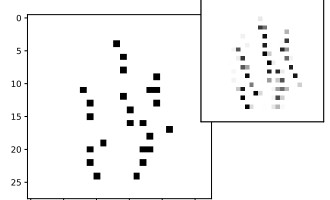

Figure 3: A realization of the learned distribution (inset) for reconstruction.

## 4.2 'LINES AND CIRCLES' IMAGE RECONSTRUCTION

**Experiment setup** To evaluate reconstruction of structured images from highly undersampled partial Fourier (k-space) measurements (keeping $3.1\%$ of the coefficients), we generated synthetic toy data comprising images that each contain up to 5 horizontal lines and randomly-sized circles. Lines and circles were placed at random positions and their pixel intensity was drawn from a uniform distribution between 1 and 10. Examples were generated in an on-line fashion during training. A

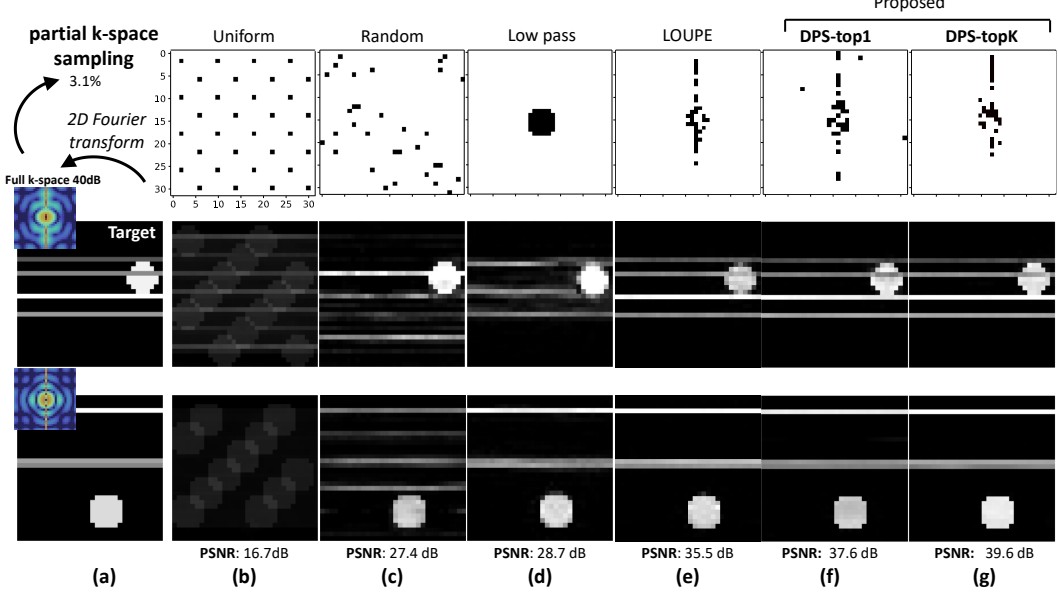

Figure 4: Image reconstruction performance from partial k-space (Fourier) measurements on a custom toy dataset consisting of lines and circles with random locations and sizes. Illustrative examples of the k-space and target images are given in (a). The sampling patterns, reconstructed images and PSNR value (across the entire test set) for the different sampling strategies are displayed in: (b) uniform, (c) random, (d) low pass, (e) LOUPE, (f) DPS-top1, and (g) DPS-topK, respectively. In all cases, only 3.1% of the Fourier coefficients have been selected.

pre-generated hold-out test set of 1000 randomly generated examples was used for all cases. Two illustrative test examples, along with their Fourier-domain representations, are given in Figure 4(a,b). We compare the results to those obtained using three fixed partial Fourier measurement baselines, following uniform, random, and low pass subsampling patterns, respectively, and the data-driven LOUPE method.

**Task model**    Image reconstruction from partial Fourier measurements was performed by following the methodology in Zhu et al. (2018). We use a deep neural network consisting of two subsequent fully-connected layers with $tanh$ activations that map the $2N$ (zero-filled) Fourier coefficients (stacked real and imaginary values) to a vector of length $N$. This vector was subsequently reshaped into a $\sqrt{N} \times \sqrt{N}$ image, and processed by 3 convolutional layers. Table 2 in appendix C provides the layer parameters of this task network.

**Training details**    The optimization problem is the same as in eq. (9), however with the loss $\mathcal{L}_s$ now defined as the negative log-likelihood:

$$\mathcal{L}_s = \mathbb{E}_{(\mathbf{x},\mathbf{s}) \sim p_{\mathcal{D}}} \|f_\theta(\mathbf{A}_{(\mathbf{\Phi})}\mathbf{x}) - \mathbf{s}\|_2^2. \tag{10}$$

The learning rates for $\mathbf{\Phi}$ and $\theta$ were $1e-3$ and $1e-4$, respectively, and $\mu$ and $\tau$ were respectively set to $2e-4$ and $5.0$. The ADAM optimizer (with settings as provided in section 4.1) was used to train in mini-batches of 128 examples. Training was stopped when the validation loss plateaued.

**Results**    An overview of the results is given in fig. 4. As expected, uniform subsampling leads to strong spatial aliasing that can not be recovered by the task model due to violation of the Nyquist criterion. In turn, random subsampling introduces an incoherent aliasing pattern, that can only partly be recovered. Although not suffering from aliasing, low pass sampling deteriorates resolution, of which the effect is particularly evident for the broadband/sharp horizontal lines. In contrast, data-driven sampling methods (LOUPE, DPS-top1, and DPS-topK) prevent aliasing by learning an appropriate sampling pattern. DPS sampling specifically, yields high-resolution accurate reconstructions with an improved PSNR evaluated on the entire test set compared to the baselines. Note how the learned

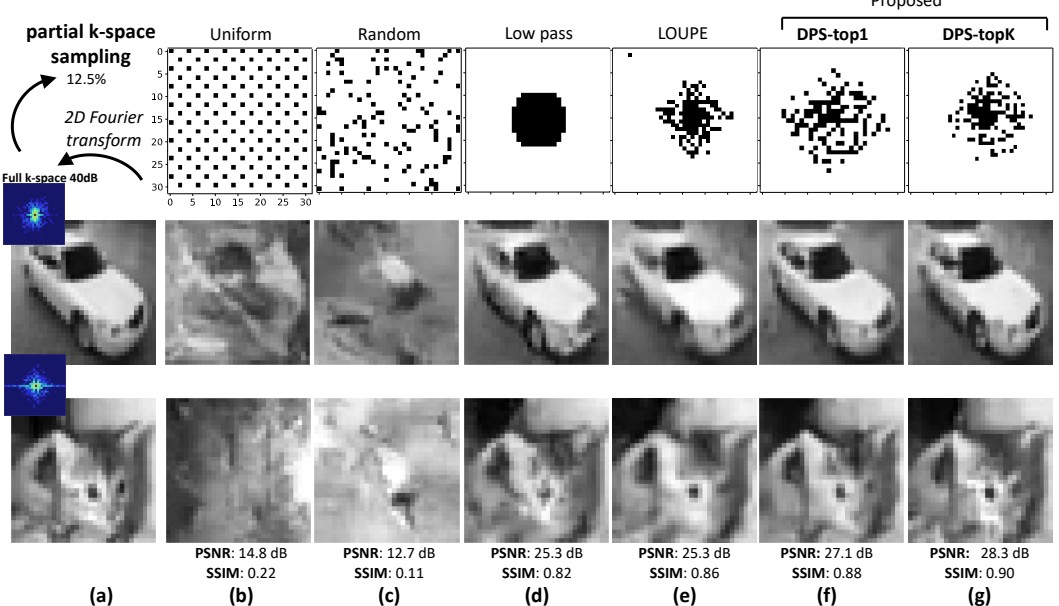

Figure 5: Image reconstruction performance from partial k-space (Fourier) measurements on the CIFAR10 database. Illustrative examples of the k-space and target images are given in (a). The sampling patterns, images and statistical quality metrics for uniform, random, low pass, LOUPE, DPS-top1, and DPS-topK are given in (b-g), respectively. In all cases, 12.5% of the Fourier coefficients have been selected.

sampling patterns (top row) have evolved from a random initialization, similar to that of (c), to the ultimate task-adaptive scheme (e-g).

## 4.3 CIFAR10 IMAGE RECONSTRUCTION

**Experiment setup**     The CIFAR10 database (Krizhevsky et al., 2009) contains 60,000 images of $32 \times 32$ pixels in 10 different classes. We converted all images to grayscale, and subsequently split them into 50,000 training images, 5,000 validation and 5,000 test images. We again learn partial Fourier sampling and image reconstruction, keeping only 12.5% of the Fourier coefficients, and compare ourselves to the three fixed partial Fourier measurement baselines and LOUPE, as described in section 4.2.

**Task model**     The challenging reconstruction task for CIFAR10 motivates the adoption of a structured model to enable strong reconstruction. We draw inspiration from iterative proximal-gradient schemes (Parikh et al., 2014), which are dedicated to solving the ill-posed linear measurement problem in eq. (1). To that end, we unfold $K = 5$ such iterations, learning an adequate image-domain proximal mapping $\mathcal{P}_\theta^{(k)}$ and stepsize $\alpha^{(k)}$ at each fold:

$$\hat{\mathbf{s}}^{(k+1)} = \mathcal{P}_\theta^{(k)} \left( \hat{\mathbf{s}}^{(k)} - \alpha^{(k)} \mathbf{F}^H \mathbf{A}_{(\boldsymbol{\Phi})}^T \left( \mathbf{A}_{(\boldsymbol{\Phi})} \mathbf{F} \hat{\mathbf{s}}^{(k)} - \mathbf{A}_{(\boldsymbol{\Phi})} \mathbf{x} \right) \right), \qquad (11)$$

where $\mathbf{F} \in \mathbb{C}^{N \times N}$ is a discrete Fourier transform (DFT) matrix, and $(\cdot)^H$ denotes the Hermitian (conjugate transpose). In the above formulation, at each fold a step is taken towards the sampling-consistent subspace that adequately represents the physical measurement of $\mathbf{s}$ by $\mathbf{A}_{(\boldsymbol{\Phi})} \mathbf{F}$. The trained proximal operator $\mathcal{P}_\theta^{(k)}$, a 7-layer convolutional network, then projects this onto the manifold of visually plausible images (Mardani et al., 2018), removing noise, aliasing, or blurring artifacts. Table 3 in appendix C provides provides details on the layers in the task model.

**Training details**     Optimization settings were similar to those in section 4.2, leveraging a mean-squared-error (negative log-likelihood) reconstruction cost and a distribution entropy penalty on $\boldsymbol{\pi}_m$. To promote visually plausible reconstructions, we added an adversarial (Ledig et al., 2017) cost by

adopting a discriminator network $D_\psi(\cdot)$ that aims to discriminate between images reconstructed from partial Fourier measurements ($\hat{\mathbf{s}}$) and actual images $\mathbf{s}$. The discriminator comprised 3 convolutional layers with leaky ReLU activations, followed by global average pooling, $40\%$ dropout, and a logistic binary classification model (parameters are provided in table 3, appendix C). The sampling- and task model parameters were then trained to both minimize the negative log-likelihood and discriminator loss $\mathcal{L}_{D_\psi}$, in addition to the entropy penalty $\mathcal{L}_e$:

$$\mathcal{L}_{D_\psi} = \mathbb{E}_{\mathbf{s},\hat{\mathbf{s}}}\left[\log(D_\psi(\mathbf{s})) + \log(1 - D_\psi(\hat{\mathbf{s}}))\right], \tag{12}$$

$$\hat{\mathbf{\Phi}}, \hat{\theta}, \hat{\psi} = \operatorname*{argmin}_{\mathbf{\Phi},\theta} \operatorname*{argmax}_{\psi}\left\{\mathbb{E}_{(\mathbf{x},\mathbf{s})\sim p_\mathcal{D}}\left[\|f_\theta(\mathbf{A}_{(\mathbf{\Phi})}\mathbf{x}) - \mathbf{s}\|_2^2\right] + \mu\mathcal{L}_e + \lambda\mathcal{L}_{D_\psi}\right\}, \tag{13}$$

where $\lambda$ weighs adherence to the data-driven MSE loss and the discriminator loss. It was empirically set to 0.004. The learning rates for $\{\mathbf{\Phi}, \psi\}$ and $\theta$ were $1e-3$ and $2e-4$, respectively, and $\mu$ was empirically set to $1e-6$, balancing sharpness and high frequency hallucinations. Temperature parameter $\tau$ was set at a constant value of 2.0. Training was performed using the ADAM optimizer (with settings as given in section 4.1) in batches of 8 images, until the validation loss plateaued.

**Results**    Figure 5 shows how task-adaptive sampling significantly outperforms all baselines. Both the uniform and random subsampling schemes suffer from severe aliasing which the task model is not able to adequately restore. While low pass sampling does not lead to aliasing, the absence of real high-frequency information causes the task model to 'hallucinate' such (super-resolution) content. Interestingly, LOUPE (e) learned a rather compact, i.e. around DC frequencies, sampling scheme, resulting in blurry images, but DPS learned a more wide-spread pattern, also capturing higher frequencies. The result of this is particularly notable for the example displayed in the bottom row of fig. 5, being less structured (or predictable) than that in the row above it. DPS thus enabled high-quality, high-resolution reconstructions that go beyond those obtained with the other methods, reaching a PSNR of 27.1 dB and 28.3 dB, and a structural similarity index (SSIM) of 0.88 and 0.90 across the 5000 test examples, for topK and top1 respectively. For the most competitive methods, i.e. fixed sampling low pass and data-driven LOUPE, these statistics were 25.3 dB and 0.82, and 25.3 dB and 0.86, respectively.

## 5  CONCLUSIONS

We have introduced Deep Probabilistic Subsampling (DPS), a framework that enables jointly learning a data- and task-driven sampling pattern, with a subsequent task-performing model. The framework is generic and can be combined with any network architecture that performs the required task using the subsampled set of signal elements. Empirically we find the method to perform strongly on toy datasets and canonical deep learning problems. Further work can explore the effectiveness of DPS in real-world problems such as MRI scanning. Even though LOUPE (Bahadir et al., 2019) learned similar subsampling patterns in all experiments, compared to DPS, its performance on the downstreak task showed to be lower. We argue that LOUPE probably suffers from the fact that samples in the training phase are relaxed. As such, the task network is optimized on a relaxed subsampled pattern, whereas a hard thresholded pattern is used for inference on the test set to make fair comparisons to DPS.

In comparing the parameter-restricted DPS-topK versus the more expressive DPS-top1, we find the former to perform better in our experiments. Studying the sampling distributions after training, we find that DPS-top1 exhibits more uncertainty. The insets in fig. 2 show this by the higher variance present among the 1000 realizations for DPS-top1 compared to DPS-topK. We tentatively attribute this due to difficulties with convergence. Further work will explore if the DPS-top1's more expressive parametrization can improve over DPS-topK in more complex problems that might benefit from a broader hypothesis space during training, as well as under more carefully tuned softmax temperature annealing schemes.

Our data-driven method does not explicitly need knowledge regarding the, often unknown, sparsifying basis, whereas conventional CS algorithms do require this. Like all data-driven optimized methods, a learned sampling scheme is at risk of overfitting to a small training dataset. Although we

did not observe this issue in our experiments, careful regularization might be required to ensure that this effect is minimal in such high-risk tasks. The fully-differentiable DPS framework allows for flexibility, and interesting extensions can be explored in future work. Rather than learning a fixed set of parameters for the subsampling distribution, a neural network can be used to predict the parameters instead, conditioned on contextual data or the samples acquired so far. Finally, our method currently requires the desired sampling rate to be predetermined as a hyperparameter. Future work can explore if this rate can be jointly optimized to incorporate optimization of the subsampling rate.

ACKNOWLEDGMENTS

This research was supported in part by Philips Research. It is also part of a research program Rubicon ENW 2018-3 with project number 019.183.EN.014, which is financed by the Dutch Research Council (NWO). We thank Wouter Kool and the anonymous reviewers for their valuable feedback.

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

## A  GRADIENT OF GUMBEL-SOFTMAX SAMPLING

For any $m^{\text{th}}$ pair of rows $(\mathbf{a}_m, \boldsymbol{\phi}_m)$, any $n^{\text{th}}$ element $a_{m,n}$ in $\mathbf{a}_m$ can be differentiated towards all elements in $\boldsymbol{\phi}_m$ through:

$$
\nabla_{\boldsymbol{\phi}_m} a_{m,n}
$$

$$
= \nabla_{\boldsymbol{\phi}_m} \mathbb{E}_{\mathbf{e}_m} \Big[ \text{softmax}_\tau (\mathbf{w}_{m-1} + \boldsymbol{\phi}_m + \mathbf{e}_m) \Big|_n \Big]
$$

$$
= \mathbb{E}_{\mathbf{e}_m} \Big[ \nabla_{\boldsymbol{\phi}_m} \text{softmax}_\tau (\mathbf{w}_{m-1} + \boldsymbol{\phi}_m + \mathbf{e}_m) \Big|_n \Big]
$$

$$
= \mathbb{E}_{\mathbf{e}_m} \Big[ \nabla_{\boldsymbol{\phi}_m} \frac{\exp\{(\mathbf{w}_{m-1} + \boldsymbol{\phi}_m + \mathbf{e}_m)/\tau\}}{\sum_{i=1}^N \exp\{(w_{m-1,i} + \phi_{m,i} + e_{m,i})/\tau\}} \Big|_n \Big] \tag{14}
$$

Gumbel noise vector $\mathbf{e}_m$ can be reparametrized as a function of a uniform noise vector $\boldsymbol{\epsilon}_m \sim \mathcal{U}(0,1)$ i.i.d., through:

$$
\mathbf{e}_m = -\log(-\log(\boldsymbol{\epsilon}_m)). \tag{15}
$$

This allows rewriting eq. (14) into:

$$
\nabla_{\boldsymbol{\phi}_m} a_{m,n} = \mathbb{E}_{\boldsymbol{\epsilon}_m} \Big[ \nabla_{\boldsymbol{\phi}_m} \frac{\exp\{(\mathbf{w}_{m-1} + \boldsymbol{\phi}_m - \log(-\log(\boldsymbol{\epsilon}_m)))/\tau\}}{\sum_{i=1}^N \exp\{(w_{m-1,i} + \phi_{m,i} - \log(-\log(\epsilon_{m,i})))/\tau\}} \Big|_n \Big]
$$

$$
= \int_{-\infty}^{\infty} \cdots \int_{-\infty}^{\infty} \mathbf{P}\big[\boldsymbol{\epsilon}_m = [k_1, \cdots, k_N]\big] \nabla_{\boldsymbol{\phi}_m} \frac{\exp\{(\mathbf{w}_{m-1} + \boldsymbol{\phi}_m - \log(-\log(\boldsymbol{k})))/\tau\}}{\sum_{i=1}^N \exp\{(w_{m-1,i} + \phi_{m,i} - \log(-\log(k_i)))/\tau\}}\Big|_n \, dk_N \cdots dk_1
$$

$$
= \int_0^1 \cdots \int_0^1 \mathbf{P}\big[\epsilon_{m,1} = k_1\big] \mathbf{P}\big[\epsilon_{m,2} = k_2\big] \cdots \mathbf{P}\big[\epsilon_{m,N} = k_N\big] \cdot
$$

$$
\nabla_{\boldsymbol{\phi}_m} \frac{\exp\{(\mathbf{w}_{m-1} + \boldsymbol{\phi}_m - \log(-\log(\boldsymbol{k})))/\tau\}}{\sum_{i=1}^N \exp\{(w_{m-1,i} + \phi_{m,i} - \log(-\log(k_i)))/\tau\}}\Big|_n \, dk_N \cdots dk_1
$$

$$
= \int_0^1 \cdots \int_0^1 1 \cdot \nabla_{\boldsymbol{\phi}_m} \frac{\exp\{(\mathbf{w}_{m-1} + \boldsymbol{\phi}_m - \log(-\log(\boldsymbol{k})))/\tau\}}{\sum_{i=1}^N \exp\{(w_{m-1,i} + \phi_{m,i} - \log(-\log(k_i)))/\tau\}}\Big|_n \, dk_N \cdots dk_1.
$$

$$
\tag{16}
$$

# B    ALGORITHM DESCRIPTION

---

**Algorithm 1** Deep Probabilistic Subsampling (DPS)

---

**Require:** Training dataset $\mathcal{D}$, Number of iterations $n_{\text{iter}}$, temperature parameter $\tau$, initialized trainable parameters $\mathbf{\Phi}$ and $\theta$.
**Ensure:** Trained logits matrix $\mathbf{\Phi}$ and task network parameters $\theta$.
    **for** $i = 1$ to $n_{\text{iter}}$ **do**
      - Draw mini-batches $(\mathbf{x}_i, \mathbf{s}_i)$: a random subset of $\mathcal{D}$
      - Compute $\mathbf{A_\Phi} = [\mathbf{a}_1; \dots; \mathbf{a}_M]$ using:
      **if DPS-top1 then**
        - Initialize mask: $\mathbf{w}_0 = \mathbf{0}$
        **for** $m = 1$ to $M$ **do**
          - Draw i.i.d. Gumbel noise samples $\mathbf{e}_m \in \mathbb{R}^N$
          - Sample from the distribution : $\tilde{r}_m = \underset{n}{\arg\max}\big\{w_{m-1,n} + \phi_{m,n} + e_{m,n}\big\}$
          - Create one-hot vector: $\mathbf{a}_m = \text{one\_hot}_N(\tilde{r}_m)$
          - Take current mask: $\mathbf{w}_m = \mathbf{w}_{m-1}$
          - Update mask: $w_{m,\tilde{r}_m} = -\infty$
        **end for**
      **else if DPS-topK then**
        - Draw i.i.d. Gumbel noise samples $\mathbf{e}_1 \in \mathbb{R}^N$
        - Sample M samples from the distribution and create an M-hot vector:
        M-hot $= \underset{n}{\text{TopK}}\big\{\phi_{1,n} + e_{1,n}\big\}$
        - Expand the M-hot vector in $M$ one-hot vectors $\mathbf{a}_m$, with $m \in \{1, \dots, M\}$
        - Create mask per sample: $\mathbf{w}_m = -\infty \cdot \mathbf{a}_m, \forall m \in \{1, \dots, M\}$
        - Create the cumulative masks $\mathbf{w}_m = \sum_{i=1}^{m} \mathbf{w}_i, \forall m \in \{1, \dots, M\}$.
      **end if**
      - Subsample the signal: $\mathbf{y}_i = \mathbf{A_\Phi}\mathbf{x}_i$
      - Compute the task: $\hat{\mathbf{s}}_i = f_\theta(\mathbf{y}_i)$
      - Compute loss using : $\mathcal{L}_i = \|\mathbf{s}_i - \hat{\mathbf{s}}_i\|_2^2 + \mathcal{L}_{\text{pen}}$
      - $\forall m \in \{1, \dots, M\}$, redefine $\nabla_{\boldsymbol{\phi}_m}\mathbf{a}_m := \nabla_{\boldsymbol{\phi}_m}\mathbb{E}_{\mathbf{e}_m}\big[\text{softmax}_\tau(\mathbf{w}_{m-1} + \boldsymbol{\phi}_m + \mathbf{e}_m)\big]$,
      where $\boldsymbol{\phi}_m = \boldsymbol{\phi}_1$ for DPS-topK
      - Use Adam optimizer to update $\mathbf{\Phi}$ and $\theta$
    **end for**

---

## C  HYPERPARAMETERS OF ADOPTED TASK MODELS

This appendix contains tables that provide details on the adopted task model architectures for the different experiments. We use the abbreviation FC, which stands for 'fully-connected layer'.

Table 1: The layer parameters corresponding to the task model in MNIST classification.

| Layer | Output size | Init | Activation |
|-------|-------------|------|------------|
| FC 1 | N | Glorot uniform[3] | LeakyReLU (0.2) |
| Dropout 1 (20%) | N | - | - |
| FC 2 | 256 | Glorot uniform | LeakyReLU (0.2) |
| Dropout 2 (20%) | 256 | - | - |
| FC 3 | 128 | Glorot uniform | LeakyReLU (0.2) |
| Dropout 3 (20%) | 128 | - | - |
| FC 4 | 128 | Glorot uniform | LeakyReLU (0.2) |
| FC 5 | 10 | Glorot uniform | Softmax |

Table 2: The layer parameters corresponding to the task model in 'lines and circles' reconstruction.

| Layer | Output size | Init | Activation | Kernel size | Strides |
|-------|-------------|------|------------|-------------|---------|
| FC 1 | N | Glorot uniform | tanh | - | - |
| FC 2 | N | Glorot uniform | tanh | - | - |
| Reshape | $\sqrt{N} \times \sqrt{N}$ | - | - | - | - |
| Conv2D 1 | $\sqrt{N} \times \sqrt{N} \times 64$ | Glorot uniform | ReLU | $5 \times 5$ | $1 \times 1$ |
| Conv2D 2 | $\sqrt{N} \times \sqrt{N} \times 64$ | Glorot uniform | ReLU | $5 \times 5$ | $1 \times 1$ |
| Conv2D 3 | $\sqrt{N} \times \sqrt{N} \times 1$ | Glorot uniform | None | $7 \times 7$ | $1 \times 1$ |

---

[3]Glorot & Bengio (2010)

Table 3: The layer parameters corresponding to the task model in CIFAR10 reconstruction.

| Network type | Layer | Output size | Init | Activation | Kernel size | Strides |
|---|---|---|---|---|---|---|
| Generator | | | | | | |
| | Proximal operator $\mathcal{P}_\theta^{(k)}$ | | | | | |
| | $6\times$ Conv2D | $\sqrt{N} \times \sqrt{N} \times 64$ | Glorot uniform | ReLU | $3\times3$ | $1 \times 1$ |
| | $1\times$ Conv2D | $\sqrt{N} \times \sqrt{N} \times 1$ | Glorot uniform | ReLU | $3\times3$ | $1 \times 1$ |
| | Stepsize $\alpha^{(k)}$ | | | | | |
| | $1\times$ Conv2D | $\sqrt{N} \times \sqrt{N} \times 1$ | Glorot uniform | ReLU | $3\times3$ | $1 \times 1$ |
| Discriminator $D_\psi$ | | | | | | |
| | Conv2D 1 | $\frac{\sqrt{N}}{2} \times \frac{\sqrt{N}}{2} \times 128$ | Glorot uniform | LeakyReLU (0.2) | $3\times3$ | $2 \times 2$ |
| | Conv2D 2 | $\frac{\sqrt{N}}{4} \times \frac{\sqrt{N}}{4} \times 128$ | Glorot uniform | LeakyReLU (0.2) | $3\times3$ | $2 \times 2$ |
| | Conv2D 3 | $\frac{\sqrt{N}}{8} \times \frac{\sqrt{N}}{8} \times 128$ | Glorot uniform | LeakyReLU (0.2) | $3\times3$ | $2 \times 2$ |
| | GlobalAveragePooling2D | 128 | - | - | - | - |
| | Dropout 40% | 128 | - | - | - | - |
| | FC 1 | 1 | - | Sigmoid | - | - |

## D  EXAMPLE TRAINING GRAPH FOR TRAINING OF 'LINES AND CIRCLES' RECONSTRUCTION USING DPS-TOPK

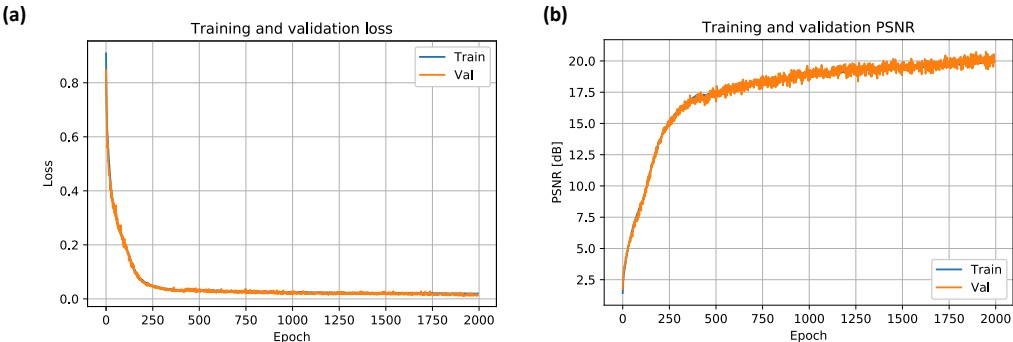

Figure 6: Training graphs for the 'lines and circles' reconstruction problem from section 4.2. (a) The graph shows that even though neural network loss functions are typically non-convex, the optimization trajectory of the loss value still followed a smooth path. The train and validation loss exactly behave the same way, suggesting no overfitting. (b) The corresponding PSNR value during training.

