# OpenReview forum: "Deep probabilistic subsampling for task-adaptive compressed sensing"
_ICLR.cc/2020/Conference — Accept (Poster)_

### Official Review · AnonReviewer3 · 2019-10-20
**Official Blind Review #3**

**Rating:** 6

**Review:**

The paper proposes a learning-based adaptive compressed sensing framework in which both the sampling and the task functions (e.g., classification) are learned jointly end-to-end. The main contribution includes using the Gumbel-softmax trick to relax categorical distributions and use back-propagation to estimate the gradient jointly with the tas neural network. The proposed solution has the flexibility of able to be used in several different tasks, such as inverse problems ( super-resolution or image completion) or classification tasks.  The paper is very well written.

The paper locates itself well in current baselines and explains Experiments mostly well. However, there are significant limitations in demonstrating the effectiveness/impact of the proposed technique:
1) The only comparison to another non-fixed sampling baseline is Kool et al. 2019. The visualization and a thorough comparison were missing in MNIST classification. This baseline was also missing in image reconstruction.
2) Compressive Sensing incorporates vast literature of algorithms focusing on different aspects of improvements; algorithms focused on classification and inverse problems. Even if done disjointly, how does the proposed joint learning is compared to those algorithms in these domains?
3) Top row of Figure 3 nicely explains how the learned sampling paradigm performs compared to other mechanisms (such as uniform, random, low-pass). But there is no comparision against other non-fixed techniques.

**Experience Assessment:**

I have published one or two papers in this area.

**Review Assessment: Checking Correctness Of Derivations And Theory:**

I assessed the sensibility of the derivations and theory.

**Review Assessment: Checking Correctness Of Experiments:**

I assessed the sensibility of the experiments.

**Review Assessment: Thoroughness In Paper Reading:**

I read the paper at least twice and used my best judgement in assessing the paper.

---

> ### Author Response · Authors · 2019-11-12
> **Reply to review #3**
>
> We thank the reviewer for the positive and constructive feedback. Below we answer the questions and concerns:
>
> Question 1:
> We agree with the referee and will therefore include a visualization of the trained distributions using Gumbel top-k sampling and a realization of the sampling pattern. We are currently running experiments to obtain Gumbel top-k results for the ‘lines and circles’ and CIFAR10 experiments as well.
>
> Since we did not sufficiently emphasize that leveraging Gumbel top-k sampling for learning signal subsampling matrices is part of the novelty of the present work, we clarified this in the revised manuscript. In fact, using Gumbel top-k sampling in this context can be seen as a constrained version of DPS, with shared weights across the M distributions.
>
> To also include previously-published baselines, we are currently running experiments with the recently proposed LOUPE method by Bahadir et al. (2019).
>
> Question 2:
> Indeed, the notion of compressed sensing has spurred vast work, ranging from sensing strategies to signal recovery algorithms. On the sensing side, sampling strategies are typically designed to satisfy the Restricted Isometry Property (RIP); describing isometry of the sensing matrix given K-sparse vectors, and thereby providing signal recovery guarantees, given an appropriate algorithm. On the algorithm side, sparsity in some basis transform is typically exploited, leveraging a wide variety of optimization algorithms spanning from proximal gradient methods to projection-over-convex-set and greedy algorithms. More recently, deep learning methods have been proposed for fast signal recovery from CS measurements, yielding state-of-the-art results.
>
> In this context, DPS adopts current practices in data-driven CS recovery, but extends this to incorporate subsampling (the sensing) in an end-to-end pipeline. Such an end-to-end (sampling-to-any-task) learning strategy opens up opportunities for data-driven optimization of sensing strategies beyond theoretically-established results.
>
> As pointed out by the referee, the shortcomings of disjoint optimization in classical CS are perhaps most evident when high-level tasks such as classification are part of the pipeline.
> As such, we are currently running additional experiments to better illustrate this.
>
> Question 3:
> We agree with the reviewer that such a comparison might be of interest.
>
> As such, we are currently running additional experiments to include a comparison to Gumbel top-k (as we did for the MNIST classification case) as well as the method proposed by Bahadir et al. (2019). Notably, and unlike our method, the latter approach does not permit setting a specific subsampling rate, with this rate is only being indirectly controlled via hyperparameter settings.

---

> > ### Author Response · Authors · 2019-11-15
> > **Further clarification about experiment regarding disjoint optimization**
> >
> > As a follow-up on our answer regarding the second question, we would like to mention that we added a case in the MNIST classification experiment (DPS-topk), in which we jointly train a reconstruction network with a subsampling pattern. We subsequently train the classifier network on the reconstructed images. It shows that learning a task-adaptive (classification in this case) sampling pattern outperforms disjoint learning of sampling and the task.

---

### Official Review · AnonReviewer2 · 2019-10-23
**Official Blind Review #2**

**Rating:** 6

**Review:**

The authors propose a new approach of deep probabilistic subsampling for compressed sensing, based on Gumbel-softmax, which is interesting.

A few points should be clarified:

- in compressed sensing one has e.g. the restricted isometry property (RIP) related to recovery. How does the new method relate to such theoretical results? Are the results and findings along similar lines as (classical) compressed sensing theory?

- Methods in compressed sensing are typically convex, e.g. using l1-regularization. What are the drawbacks of using deep learning in this context, e.g. related to non-convexity? What is the role of initialization?

- Does the method both work for underdetermined and overdetermined problems (number of data versus number of unknowns)?

- What is the influence of the hyper-parameters mu and lambda in eq (14)? How should the model selection be done (currently lambda is set to 0.004 without further motivation)?

- MNIST: 60,000 instead of 70,000?


**Experience Assessment:**

I have published one or two papers in this area.

**Review Assessment: Checking Correctness Of Derivations And Theory:**

I assessed the sensibility of the derivations and theory.

**Review Assessment: Checking Correctness Of Experiments:**

I assessed the sensibility of the experiments.

**Review Assessment: Thoroughness In Paper Reading:**

I read the paper at least twice and used my best judgement in assessing the paper.

---

> ### Author Response · Authors · 2019-11-12
> **Reply to review #2 (part 1)**
>
> We thank the reviewer for the interesting questions regarding our work. Below we respond to these questions of the referee:
>
> Question 1:
> In compressed sensing, RIP is indeed used to provide a measure for isometry when “restricted” to k columns, i.e. given a measurement $\mathbf{y}=\mathbf{A}\mathbf{x}$, of a k-sparse vector $\mathbf{x}$. For many practical problems of interest, analysis of measurements of sparse vectors is achieved by reformulating the measurement as $\mathbf{y}=\mathbf{A}\mathbf{\Psi}\mathbf{x}$, with $\mathbf{\Psi}$ being a sparsifying basis, and $\mathbf{z}=\mathbf{\Psi}\mathbf{x}$ being the quantity of interest. Then, RIP should be evaluated for $\mathbf{A}\mathbf{\Psi}$. A common requirement posed for sparse bases is therefore incoherence of the columns.
>
> Instead, we here directly learn a mapping to $\mathbf{z}$ from data, with no explicit notion of such a sparsifying basis. While this makes theoretical assessment more challenging, it alleviates the need for manual identification of a proper sparse basis for each new problem.
>
> We augmented parts of the discussion of the revised manuscript to better reflect this.
>
> Question 2:
> The reviewer raises a fundamental and interesting question regarding the typical loss surface in CS compared to the one of a neural network. Indeed the loss surface of a NN is highly non-linear and non-convex, it typically contains a vast amount of local minima, as a consequence of the weight space symmetry property (Goodfellow et al., 2016), i.e. having the same loss value for a different ordering of the same weights. The size of the gap between local and the global minima remains an open field of research. However, citing from Goodfellow et al. (2016):
> “The problem remains an active area of research, but experts now suspect that, for suﬃciently large neural networks, most local minima have a low cost function value, and that it is not important to ﬁnd a true global minimum rather than to ﬁnd a point in parameter space that has low but not minimal cost (Saxe et al., 2013; Dauphin et al., 2014; Goodfellow et al., 2015; Choromanska et al., 2014)”
>
> As such, we leverage the empirically-shown ability of stochastic gradient descent to optimize this non-convex function. Indeed there are no global convergence guarantees, but we have the strong advantage compared to typical L1-reconstruction algorithms that we do not need explicit knowledge on the, in practice often unknown, sparsifying basis.
>
> We followed standard practice in deep learning by  initializing all layers with their default Keras initializations, i.e. glorot uniform (Glorot, 2010), which we found to be working well. The logits of the distributions to be trained in the subsampling part of the network were initialized as a uniform, i.e. high-entropy, distribution, enabling most freedom for explorability of the sampling pattern by not explicitly setting a prior.
>
> We now detail on the initializations for each layer in the appendix of the revised manuscript.
>
> Question 3:
> For linear problems such as reconstruction, there is a clear relationship between the number of input data versus number of unknowns. In this paper we focus on non-linear reconstruction, as well as other tasks such as object classification. Under this scope, it becomes unclear if we can still consider problems to be over- or underdetermined from a traditional point of view, and a more general information theoretic standpoint might proof fruitful. Concretely, as deep learning methods are optimized stochastically, they are expected to be drawn towards solutions that carry the largest signal for the downstream task. In the face of redundant input samples and under pressure of a small number of output samples, the method is thus expected to randomly select just one of these redundant samples as this would improve the loss of the model. As some tentative evidence to support this claim, we would refer you to figure 2a (96.8% removed), where almost no directly neighbouring pixels are sampled, showing a clear preference of the model for skipping redundant samples.

---

> ### Author Response · Authors · 2019-11-12
> **Reply to review #2 (part 2)**
>
>
> Question 4:
> We thank the reviewer for pointing this out. The appropriate setting of these parameters is indeed important and we are happy to explain its impact on our outcomes.
>
> Lambda is weighing the adherence to the data-consistency (via MSE error) and visual plausible images (via the cross-entropy discriminator loss). For higher lambda, the discriminator loss is weighted more heavily, and as such, more effort is done by the model to create visually plausible natural images, as they appear in the CIFAR10 database. A balance should be found here, as a too high lambda can cause image predictions that look rather natural, however do not resemble the target image.
>
> The mu parameter weights the influence of the entropy penalty during training of the logits in the categorical distributions. For higher mu, the distributions converge quicker towards low entropy distributions, i.e. one class has a probability close to 1, as high entropy is highly penalized. There is a tradeoff here between quick convergence and explorability. When the distributions converge really quickly, the model has no chance to explore different subsampling patterns and as such will probably find subsampling patterns that are not useful
> for the downstream task. However, setting mu too low, can easily slow down convergence of the whole model.
>
> As suggested by the referee, we will detail on the above in our revised manuscript.
>
> Question 5:
> Thanks for pointing out this typo, 70,000 is the total number of datapoints that we used, of which 60,000 were used for training. We corrected this in the revised paper.

---

### Official Review · AnonReviewer1 · 2019-10-24
**Official Blind Review #1**

**Rating:** 6

**Review:**

This paper introduces  a novel DPS(Deep Probabilistic Subsampling) framework for the task-adaptive  subsampling case, which attempts to resolve the issue of end-to-end optimization of an optimal subset of signal with jointly learning a sub-Nyquist sampling scheme and a predictive model for downstream tasks. The parameterization is used to simplify the subsampling distribution and ensure an expressive yet tractable distribution.  The new approach contribution is applied to  both reconstruction and classification tasks and demonstrated with a suite of experiments in a toy dataset, MINIST, and COFAR10.


Overall, the paper requires significant improvement.

1. The approach is not well justified either by theory or practice. There is no experiment clearly shows convincing evidence of the correctness of the proposed approach or its utility compared to existing approaches (Xie & Ermon (2019); Kool et al. (2019); PlÂšotz & Roth (2018) ).

2. The paper never clearly demonstrates the problem they are trying to solve (nor well differentiates it from the compressed sensing problem  or sample selection problem)

   The method is difficult to understand, missing many details and essential explanation, and generally does not support a significant contribution.

3. The paper is not nicely written or rather easy to follow. The model is not well motivated and the optimization algorithm is also not well described.

4. A theoretical analysis of the convergence of the optimization algorithm could be needed.

5. The paper is imprecise and unpolished and the presentation needs improvement.

**There are so many missing details or questions to answer**

1. What is the Gumbel-max trick?
2. How to tune the parameters discussed in training details in the experiments?
3. Why to use experience replay for the linear experiments?
4. Are there evaluations on the utility of proposed compared to existing approaches?
5. Does the proposed approach work in real-world problems?
6. Was there any concrete theoretical guarantee to ensure the convergence of the algorithm.

[Post Review after discussion]: The uploaded version has significantly improved over the first submission. It is now acceptable.


**Experience Assessment:**

I have published one or two papers in this area.

**Review Assessment: Checking Correctness Of Derivations And Theory:**

I assessed the sensibility of the derivations and theory.

**Review Assessment: Checking Correctness Of Experiments:**

I assessed the sensibility of the experiments.

**Review Assessment: Thoroughness In Paper Reading:**

I read the paper at least twice and used my best judgement in assessing the paper.

---

> ### Author Response · Authors · 2019-11-12
> **Reply to review #1 (part 2)**
>
>
> Question 3:
> We did our best to write the paper such that it includes all details needed to fully understand the proposed method and its theoretical background. The referee indicates (in sharp contrast to referee 3) that the paper is not nicely written, nor easy to follow. We invite the referee to be specific about the sections of the original manuscript that need more clarification, allowing us to revise these sections.
>
> Question 4:
> The optimization algorithm used is the ADAM optimizer (Kingma & Ba, 2014). We refer to section 4 of Kingma & Ba (2014) for a proof of convergence for convex functions. It is known that that loss surfaces of deep neural networks are typically non-convex, however the gap between global and local minima is believed to be small for (see our answer to the referee’s last question for more detail on this statement).
>
> Question 5:
> We kindly ask the reviewer to elaborate on the given statement. Could the reviewer indicate which sections are found to be imprecise and unpolished, and which parts of the manuscript need a better presentation?
>
> =====================================
> Second part of the review
> =====================================
> Question 1:
> We specify the Gumbel-max trick in the paragraph below equation 4. To make the paper more self-contained, we will extend this paragraph to further clarify the Gumbel-max trick.
> We also refer to our answer to the first question of this referee, in which we elaborated more on the Gumbel-max trick as well.
>
> Question 2:
> All training parameters were tuned empirically. However, we agree it is worth elaborating on our insights regarding the influence on performance of some of them. We experienced that performance was most sensitive to the learning rates for the sampling and task models, and the temperature parameter tau of the softmax relaxation. We augmented the discussion of our revised manuscript to share these insights. .
>
> Question 3:
> We know experience replay as a reinforcement learning technique for storing previous state/action pairs. However, our method does not make use of reinforcement learning, so could the reviewer please elaborate how experience replay would relate to our method?
>
> Question 4:
> In Section 4.1 (MNIST classification) we already compared our proposed sampling method to used Gumbel top-K sampling for data subsampling. We are currently also running experiments that allow for extensive comparison with the recently proposed LOUPE method by Bahadir et al. (2019).
>
> Question 5:
> A large part of the experiments in this work are focusing on compressive/partial Fourier measurements. This adequately reflects the measurement setup in many real-world problems, such as k-space measurement in magnetic resonance imaging (Lustig et al.), Xampling for ultrasound imaging (Eldar et al.), and non-uniform step frequency radar (Huang, 2014). In addition, we cover direct pixel sampling, related to real-world applications such as compressive cameras. We would like to emphasize that the proposed approach is measurement-domain agnostic, and therefore can be applied across a vast amount of real-world problem.
>
> In addition, our ongoing research already shows promising results for real-world applications such as magnetic resonance imaging and ultrasound imaging. This is part of future work.
>
> Question 6:
> The trends towards using deep learning for data-driven compressed sensing indeed has the downside of not having guarantees on finding a global minimum, as the loss surface of a NN is highly non-linear and non-convex. Still, these data-driven results have shown to be very promising (Gregor et al., 2010; Jin,2019; Bahadir et al., 2019; Mousavi, 2019)
>
> However due to the weight space symmetry problem (Goodfellow et al., 2016) the loss surface contains a vast amount of local minima with the same error value. The size of the gap between local and the global minima remains an open field of research. However, citing from Goodfellow et al. (2016):
> “The problem remains an active area of research, but experts now suspect that,
> for suﬃciently large neural networks, most local minima have a low cost function
> value, and that it is not important to ﬁnd a true global minimum rather than to
> ﬁnd a point in parameter space that has low but not minimal cost (Saxe et al.,
> 2013; Dauphin et al., 2014; Goodfellow et al., 2015; Choromanska et al., 2014)”
>
> As such, we leverage the empirically-shown ability of stochastic gradient descent to optimize this non-convex function in our NN for finding local minima. Indeed there is no guarantee on finding a global optimum.

---

> > ### Comment · AnonReviewer1 · 2019-11-15
> > **Response**
> >
> > Thank you for the kind response and answering my questions clearly. The reviewer agrees that this paper did extensive work of deep probabilistic subsampling for compressed sensing based on the Gumbel-max trick and back-propagation.
> >
> > However, the justification of this paper should be realized with extensive comparison with other competitors on large datasets. The reviewer expects to see the extensive results in comparison with state-of-the-art compressed sensing methods in the revision.
> >
> > The reviewer also expects the authors include the explanations into revision. In particular,  for such description "the main shortcoming of compressed sensing:
> > “These [compressed sensing] methods, however, are lacking in the sense that they do not fully exploit both the underlying data distribution and information to solve the downstream task of interest.”",
> > please include evidence to support your claim, otherwise it is too subjective. Does that mean the proposed algorithm  "fully exploit both the underlying data distribution and information to solve the downstream task of interest"? If so, please also provide evidences.
> >
> > The authors' response did not  include their willing to provide an algorithm description. The review would expect to see an algorithm description. In addition, a description to Gumble-max in  the main text or the appendix is good for readers to get know the distribution form.
> >
> > Though it is hard to show the convergence rate in theory, an empirical convergence curve is better given.
> >
> > The reviewer is willing to change the score if the above concerns get addressed.

---

> > > ### Author Response · Authors · 2019-11-15
> > > **Response**
> > >
> > > Thank you for your getting back to us. We are happy to share that we have extended our experiments to include the recent competing method LOUPE from Bahadir et al. (2019). Other methods in traditional compressive sampling literature focus only on the reconstruction task and thus do not fit the full scope of our experiments. Besides, the algorithms typically used for this task, e.g. proximal gradient schemes, follow an iterative nature and are therefore less applicable for real-time applications.
> > > Lastly, we argue that the fixed random subsampling pattern followed by an unfolded proximal gradient scheme for the CIFAR10 reconstruction case follows classical CS principles, in which partial Fourier measurements are used by an iterative proximal gradient scheme in order to reconstruct the original signal. We clearly see worse performance here compared to DPS, as a random pattern is still pseudo-random and therefore often still not able to prevent aliasing artifacts.
> > >
> > > We agree that it is too strong a claim to say that DPS fully exploits the data distribution and information-need of a downstream task. We have reformulated this section to read "to focus solely on the information required to solve the downstream task given the underlying data distribution". We do not aim to claim that our method finds the global optimum in this situation, but rather focus on the empirical evidence that DPS performs well on a variety of tasks. We rely on the general accepted understanding that deep learning methods effectively discover patterns relevant to the supervised task.
> > > Though, we added a case in which we train MNIST reconstruction, followed by a separately trained classifier. This case confirms our hypothesize that task-adaptive learning is beneficial.
> > >
> > > After further tuning the parameters, and bringing DPS-topK better in line with the implementation of DPS-top1, we published new results on DPS-top1 and topK for all cases. The algorithmic description of DPS-top1 and DPS-topK is now added in appendix b as well.
> > >
> > > Finally we have extended our manuscript with a more elaborate description of the Gumbel-max trick (below eq. 4), and a definition of the probability distribution (footnote 1). Moreover, tables including all the layer parameters of the task networks (appendix c), and training curves demonstrating empirical convergence (appendix d) are added. We believe this should address all the concerns mentioned before.

---

> ### Author Response · Authors · 2019-11-12
> **Reply to review #1 (part 1)**
>
> We thank the reviewer for the feedback. The reviewer states in his/her summary: The parameterization is used to simplify the subsampling distribution. We would like to comment on this, by stating that the reparametrization is not used for simplifying the subsampling distribution; on the contrary it actually enables sampling from this trained distribution. In fact, without this reparametrization, our generative sampling model (DPS) would not be differentiable. Below we elaborate upon the questions and raised concerns by the reviewer:
>
>
> Question 1:
> We respectfully disagree with the referee’s conclusions, and will elaborate on the above statements in the following. While we disagree, we did however undertake a significant effort to further clarify and provide additional evidence in the revised manuscript, taking into account these comments.
>
> Regarding the theoretical correctness of deep probabilistic subsampling, in section 3.2 we explain how we incorporate a well-known reparametrization trick, termed the Gumbel-max trick (Gumbel,1954), to sample from a categorical probability distribution. Note that this shares similarities with the reparameterization trick used for sampling from trained gaussian distributions in a vanilla variational autoencoder. The Gumbel-max reparametrization perturbs the logits of the categorical distribution with Gumbel noise after which, by means of the argmax, the highest value is selected. Gumbel (1954) showed that this reparametrization allows sampling from the original categorical distribution.
> Recent state-of-the art work on a relaxation of this trick, termed Gumbel-softmax sampling (Jang et al., 2017) or the concrete distribution (Maddison et al., 2016), allows us to apply this relaxed reparametrization inside a neural network as it enables gradient calculation, which is needed for error backpropagation in the training procedure of the network. We would like to ask the reviewer what is believed to be missing from this explanation on the subsampling part of our proposed method.
>
> Regarding the theoretical basis used for the design of the task network; we took a theoretically principled approach by exploiting a model-driven network architecture for the CIFAR10 reconstruction problem. To that end, we unfold the iterations of a proximal gradient scheme (Mardani et al., NeurIPS, 2018), allowing for explicit embedding of the acquisition model (and therewith the learned sampling) in the reconstruction network.
>
> Regarding the referee’s conclusion that the manuscript lacks comparison to the approaches of (Xie & Ermon (2019); Kool et al. (2019); Plötz & Roth (2018): We would like to point out that these three references all together put forward the Gumbel top-k method. Note that the use of the Gumbel top-k method for compressive sampling is also new, and in fact constitutes a specific case (constrained version with shared weights across distributions) of the proposed deep probabilistic subsampling (DPS) framework. In the MNIST experiments we already included Gumbel top-k sampling, but we will also add this for the other experiments in the revised manuscript. In addition, we added a thorough comparison of the DPS to LOUPE (Bahadir et al, 2019), a recently proposed data-driven method for subsampling.
>
> Question 2:
> We would first like to refer the referee to third paragraph of the introduction, where we explicitly formulate the main shortcoming of compressed sensing:
>
> “These [compressed sensing] methods, however, are lacking in the sense that they do not fully exploit both the underlying data distribution and information to solve the downstream task of interest.”
>
> Then, in the list of main contributions, we write:
> “DPS: A new regime for task-adaptive subsampling using a novel probabilistic deep learning framework for jointly learning a sub-Nyquist sampling scheme with a predictive model for downstream tasks”
>
> Subquestion 2:
> We are of course willing to further specify any details that the referee misses in the current paper. We would therefore like to kindly invite the referee to be specific about the details that he/she would like to be added to the manuscript.
>
> We respectfully disagree with the referee’s conclusion that the method does not support a significant contribution. We propose a fully-probabilistic generative model for trainable sampling, that exploits both the underlying data distribution and information to solve the downstream task of interest. Our generative model builds upon recent advances on Gumbel max and top-k reparameterizations and their relaxations, showing for the first time how discrete sample selection can be done in a data-driven and task-adaptive fashion. This opens up a vast array of new opportunities in compressed sensing.

---

### Decision · Program_Chairs · 2019-12-19

**Decision:**

Accept (Poster)

**Comment:**

This paper introduces a probabilistic data subsampling scheme that can be optimized end-to-end.  The experimental evaluation is a bit weak, focusing mostly on toy-scale problems, and I would have liked to see a discussion of bias in the Gumbel-max gradient estimator.

It's also not clear how the free hyperparameters for this method were chosen, which makes me suspect they were tuned on the test set.

However, the overall idea is sensible, and the area seems under-explored.

---

> ### Author Response · Authors · 2020-02-14
> **Answer on final decision**
>
> We thank the referees for the final decision of accepting the paper, and the final comments.
> Regarding these final comments;
>
> We now added some explanation to section 3.2, in which we explain how the bias and the variance of the grandient's estimator depend on the temperature of the softmax in Gumbel-softmax sampling. We did not investigate this bias ourselves, however we add two references that aim to reduce the gradient's estimator bias.
>
> For all experiments, we had separate train, validation and test sets. As was already mentioned in the manuscript, training was stopped when the validation loss plateaued. We also tuned hyperparameters solely on the validation set, and only used the test set to run final inference for the results sections. In the final version, we included a link to our open source code, in which one can see as well that the test set is only used for final inference.